# Controlled filamentation instability as a scalable fabrication approach to flexible metamaterials

William Esposito[1], Louis Martin-Monier[1], Pierre-Luc Piveteau [1], Bingrui Xu[2], Daosheng Deng [3] ✉ & Fabien Sorin [1] ✉

Long and flexible arrays of nanowires find impactful applications in sensing, photonics, and energy harvesting. Conventional manufacturing relies largely on lithographic methods limited in wafer size, rigidity, and machine write time. Here, we report a scalable process to generate encapsulated flexible nanowire arrays with high aspect ratios and excellent tunable size and periodicity. Our strategy is to control nanowire self-assembly into 2D and 3D architectures via the filamentation of a textured thin film under anisotropic stretching. This is achieved by coupling soft lithography, glancing angle deposition, and thermal drawing to obtain well-ordered meters-long nanowires with diameters down to 50 nanometers. We demonstrate that the nanowire diameter and period of the array can be decoupled and manipulated independently. We propose a filamentation criterion and perform numerical simulations implementing destabilizing long-range Van der Waals interactions. Applied to high-index chalcogenide glasses, we show that this decoupling allows for tuning diffraction. Finally, harnessing Mie resonance, we demonstrate the possibility of manufacturing macroscopic meta-grating superstructures for nanophotonic applications.

Owing to their large aspect ratio and intrinsic electronic and optical properties, one-dimensional architectures such as nanowires (NW) are at the heart of innovative components in sensing, energy harvesting and photonics. Made with the right materials from III-V alloys to oxides or chalcogenide (ChG) glasses, NW arrays can be used as low power phase-change memory devices[1,2], low-loss waveguides[3] for high confinement, highly-localized near-field optical sensing[4], or form efficient optoelectronic systems[5,6]. Optical properties can also be engineered to enhance light absorption or scattering as desired[7,8], and when well-engineered, high index ordered filament arrays can enable the integration of functionalities such as lensing over distances orders of magnitude smaller than traditional methods[9].

NWs are usually obtained by cleanroom methods, such as lithographic top-down and bottom-up approaches[10–14]. Despite intrinsic

advantages such as high accuracy and repeatability, dimensions are limited to that of the wafer, with NWs of a few millimeters in length at most. Obtaining ultra-long (a few tens-of-meters) NWs organized as an array in a flexible support remains however a significant challenge. Beyond gas phase deposition approaches, fluid dynamics-based strategies have been proposed to generate longer NWs within different types of substrates and three-dimensional arrangements, such as fluid-spinning[15,16] or sonochemical growth[5]. However, none of these methods allow regular positioning and alignment of the NWs. To produce arrays, pressure-assisted melt filling[17] or chemical vapor deposition[18] inside microstructured glass optical fibers has been demonstrated within channels down to several hundred nanometers over lengths up to a few centimeters. However, the deposition time in this process scales as the square of its length, thereby limiting upscaling[19].

[1]Laboratory for Photonic and Fiber devices, Ecole Polytechnique Fédérale de Lausanne (EPFL), Lausanne, Switzerland. [2]Department of Basic Courses, Naval University of Engineering, Wuhan, China. [3]Department of Aeronautics and Astronautics, Fudan University, Shanghai, China. ✉e-mail: dsdeng@fudan.edu.cn; fabien.sorin@epfl.ch

To overcome this limitation, thermal drawing has attracted considerable attention as an alternative for fabricating directly NW arrays with very high aspect ratios[5,20–24]. Thermal drawing is a fiber manufacturing process during which a macroscopic multimaterial structure −the "preform"−is heated and stretched down to microscopic transverse dimensions. One method to manufacture ordered nanoarrays through thermal drawing relies on the stack-and-draw technique[25], whereby a single-core fiber is drawn, cut into equal sections, assembled into a bundle, and re-drawn. Obtaining regularity in fiber spacing and filament diameter depends on the precise stacking and uniformity of the previous steps. Reaching the nanoscale typically requires three successive draws[26], which often lead to intermediate preform shrinkage and rupture because of residual stress[27]. Direct thermal drawing of micron-thick films has been shown to lead to dewetting transversal to the fiber axis. Flow in the axial direction has the effect of scaling down surface perturbations, hence preventing capillary break-up and preserving continuity over the entire fiber length[28,29]. Semiconducting filaments at the nanoscale could be produced, opening exciting opportunities in scalable manufacturing of NW arrays[24]. Nevertheless, irregular transversal dewetting limits control over geometrical parameters. This is particularly detrimental to many practical applications in photonics which require a high degree of order. The possibility to reduce inter-filament spacing is also desirable, allowing for collective interaction and interference effects. Indeed, the physics of capillary break-up (i.e., the wavelength of the fastest growth mode[30]) determine large filament spacing and inherently hinder control over the wire diameter-to-period ratio, key to advanced photonic applications.

Template dewetting is another fluid-based process leading to better control over geometrical parameters of the resulting nanoarrays. It consists of the reflow of a thin viscous layer on a textured substrate that imposes dewetting at specific locations[31–33]. This approach has been recently exploited to generate ChG NWs on a rigid wafer-like substrate[34]. While the results were promising, the conditions to engineer reflow-induced dewetting faster than capillary break-up are very challenging, limiting diameter uniformity and achievable feature sizes. Moreover, the NW length is limited by the nanoimprinted area, typically of a wafer-scale. To date, the challenges associated with understanding and controlling fluid dynamics-based

processes to realize ordered NW arrays with large aspect ratios, with well controlled size and periodicity, remain unresolved.

Herein, we propose dynamic template dewetting as a novel viscous flow process to achieve unprecedented control over diameter and periodicity of ordered NW arrays, potentially kilometers in length. We propose to tune the modes of dewetting in a textured thin film of a material obtained from angle deposition on a nanoimprinted polymer substrate. This assembly is then thermally drawn, continuously triggering instabilities at prescribed positions in the fiber cross-section while preventing instabilities in the drawing direction. We demonstrate meter-long periodic arrays of ChG NWs in a single drawing step, encapsulated within a flexible polymer, and with diameters down to 50 nm. We also successfully master the ability to independently tune NW diameter and spacing, providing unprecedented control over transverse geometrical parameters. Further combining several films into a single preform, 3D stackings of 2D arrays are fabricated, enabling the integration of multiple photonic functionalities over reduced distances. Using multi-scale fluid dynamics simulations, we develop a model to account for the rearrangement process down to nanometric dimensions. The model further establishes universal conditions under which such filamentation occurs, applicable to other materials such as metallic glasses[22,35] or conductive composites[36–38]. Finally, since the versatility of the drawing process allows engineering of the NW array to fit optical requirements, we propose an architecture for applications that benefit from the high refractive index of ChG. In particular, we achieve controlled diffraction effects and the scalable fabrication of flexible 1D optical all-dielectric metamaterial, demonstrating the impact of our nano-fabrication approach for flat optics and nano-photonics.

## Results
### Description of the process

The proposed process is shown in Fig. 1 and starts with soft lithography[39] where a primary silicon master mold with line arrays of fixed periodicity $P$ obtained by photolithography (see Methods) is used for drop-casting and curing of Polydimethylsiloxane (PDMS) negative texture replicas. Such soft stamps allow for hot embossing the initial positive texture onto thermoplastic substrates. Polyetherimide (PEI) was chosen as a cladding material for its high glass

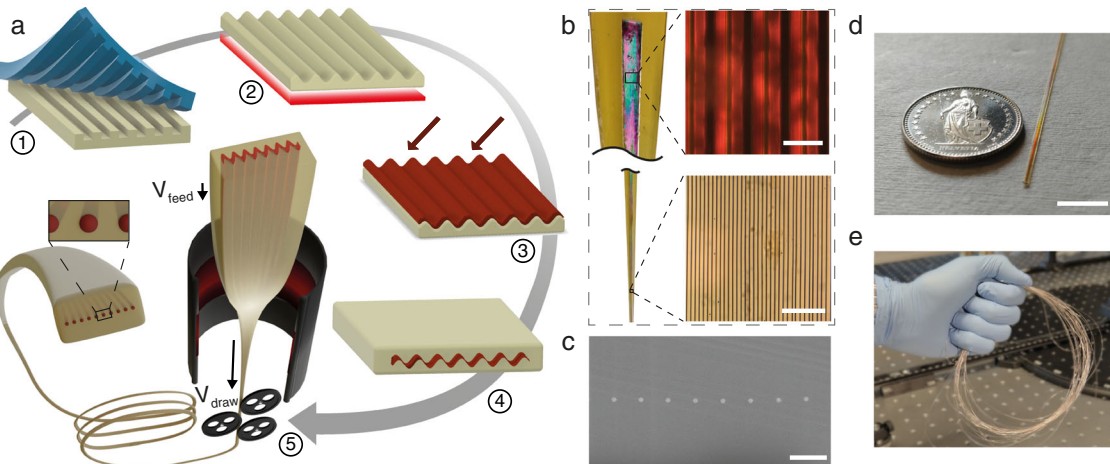

**Fig. 1 | High throughput NW array fabrication process and observation. a** 1. A microscopic square pattern with period $P$ is imprinted on a thermoplastic polymer with a PDMS stamp. 2. The polymer film is then reflowed to smooth out the pattern. 3. A ChG As$_2$Se$_3$ (or Se) is thermally evaporated onto the polymer substrate at an angle resulting in a thickness fluctuation. 4. The films are encapsulated into a "preform" by hot-embossing another polymer film on top. 5. The preform is thermally drawn to microscopic dimensions by adjusting furnace temperature, drawing, and feeding speeds. The film breaks up periodically into long ordered NWs.

**b** During thermal drawing, the continuous film in the preform (top) is heated and stretched, subjected to fluid instability and subsequently producing well-ordered filament arrays (bottom). **c** SEM cross section of a fiber thermally drawn, demonstrating the transversal breakup. **d** Close-up picture of a ChG NW array encapsulated inside a polymer fiber. **e** Meters of flexible fibers can be manufactured in a single drawing step. Scale bars are respectively: **b** 10 µm (top), 5 µm (bottom), **c** 1.4 µm, **d** 10 mm.

transition temperature $T_g$ ~ 217 °C[40], suitable for co-drawing with ChGs such as $As_2Se_3$[29,41] ($T_g$ ~ 174 °C)[42]. Polysulfone (PSU, $T_g$ ~ 143 °C)[43] has also been employed in combination with Se ($T_g$ ~ 47 °C)[21], at a lower processing temperature and higher cladding-to-film material viscosity ratio during thermal drawing. The next step involves thermal reflow of the texture, which smoothes out the square surface profile into a sinusoidal interface[44], required to avoid sub-period thickness fluctuations (Supplementary Note 1). In a third step, the ChG is thermally evaporated at an angle (Supplementary Note 2) to ensure a variation in film thickness following the substrate pattern. The deposited ChG film is then encapsulated by annealing under light pressure to ensure intimate conformity with the polymeric cladding. Finally, the preform is lowered into a furnace and thermally drawn into a fiber. During annealing and stretching (see Methods), the ChG film breaks up into an array of NWs (Fig. 1b, c) encapsulated in meters-long flexible fibers (Fig. 1d, e). The model below describes how this occurs.

## Filamentation model

During thermal drawing, the evolution of the textured film is determined by two competing phenomena[30,45]. On one hand, interfacial surface energy is minimized by texture flattening, driven by surface tension and interface curvature. On the other hand, Van der Waals (VdW) interactions are intrinsically destabilizing[46]. Any deformation of either film interface (top/bottom) from an initial situation of constant thickness $H$ is referred to as a "perturbation", which can be decomposed as a sum of modes[24]. When both interfaces exhibit symmetry along the mid-film plane, thickness remains constant along the cross-section ("sinuose" mode). Conversely, anti-symmetric perturbations correspond to thickness fluctuations ("varicose" mode)[24].

This fundamental understanding of thin film dewetting dynamics led us to propose substrate nanoimprinting and glancing angle deposition (or thermal reflow, see Supplementary Note 4) to engineer the initial film thickness at the preform level to control the dewetting path during drawing, resulting in a combination of sinuous and varicose modes termed "template instability/perturbation" (Fig. 2a). We anticipate that as the fiber is scaled down (Draw Ratio DR(t)) between insertion into the drawing furnace at time t = 0 and exiting it at t = $t_f$, sinuose modes decay while varicose modes grow, leading to film thinning and rupture at prescribed locations (Fig. 2b). However, for flat films, given the randomness of the initial amplitude distribution, filaments with a broad distribution in both size and spacing are obtained with very limited control. Our strategy is instead to select the dominant dewetting instability of the textured film (total growth rate $\Omega_T$, see Supplementary Note 6) by imposing initial perturbations at the preform level via imprinting and angle deposition to realize the preferred wavelength. As in the model for anisotropic instability of a stretching viscous sheet (AISVS)[24], we therefore need the template perturbation to (i) grow, and (ii) dominate over the perturbation of fastest growth rate $\Omega_{max}$ of a flat film. Dashed lines on Fig. 2c give minimum thicknesses required for (i), i.e. $\Omega_T > 0$, once Van der Waals forces play a pronounced role over the surface tension. Moreover, (ii) requires the templated instability amplitude $a_T(t)$ to remain larger than the fastest-growing mode amplitude $a_{max}(t)$ throughout the draw, i.e. their ratio should remain larger than 1. This translates to: $\frac{a_T(t)}{a_{max}(t)} = \frac{a_T(t=0)}{a_{max}(t=0)} R_{growth}(t) > 1$, where $R_{growth}(t) = \exp([\Omega_T - \Omega_{max}]t)$ is the relative amplitude growth during the draw. We estimate for a typical textured film: $\frac{a_T(t=0)}{a_{max}(t=0)}$ ~ 100 (Supplementary Note 6), and therefore a sufficient condition is that: $R_{growth}(t_f) > \frac{1}{100}$. In Fig. 2d, dashed lines delimit maximum thicknesses for which (ii) is fulfilled in this conservative case.

Quantitative predictions should be considered with care, as they stem from a linear analysis for small perturbations[20,23,24,47]. Nevertheless, they offer physical insight into the underlying dynamics at play. Fully modeling the non-linearized case can also be achieved by numerically solving (Navier-Stokes) fluid dynamics equations, which we now turn to.

While surface tension obviously tends to flatten a thin film by smoothing the interface to reduce the total interfacial energy, implementing the disjoining pressure in the general case of large amplitudes is comparably more challenging. Based on previous works[48], a complete non-retarded VdW force can be expressed for both the cladding and the film (Supplementary Note 7), relying essentially on the Hamaker procedure combined with the Lifschitz approach[48,49]. Multiphase computational fluid dynamics (CFD) modeling can now be used to solve (non-linear) Navier-Stokes equations implementing this body force. Using an iterative scaling algorithm, we simulate the multi-scale physics at play in the thermal drawing process. We adapt a Lagrangian specification for the flow field, following an elementary fiber unit slice during the draw (Supplementary Note 7).

To study the evolution of the film, we introduce a dimensionless height ratio $R_{height} = \frac{H_{max} - H_{min}}{H_{max} + H_{min}}$ where $H_{max}$ (resp. $H_{min}$) is the maximal (resp. minimal) thickness over a unit period. $R_{height}$ characterizes the reflow associated with the varicose mode: $R_{height}$ = 0 indicates complete reflow, while $R_{height}$ = 1 indicates film break-up. Evolution of this parameter in time is plotted in Fig. 2e, for different initial values (sensitivity to simulation parameters in Supplementary Note 8). When reaching smaller dimensions towards the end of the draw (high DR), the disjoining pressure becomes sufficiently strong to induce local thinning. $R_{height}$ grows rapidly and the divergence of the destabilizing forces beyond this point becomes challenging to capture. In these cases, the curves are extrapolated in dashed lines. Conversely, when the initial varicose mode amplitude was lower ($R_{height,0} \le 0.6$), the film remains continuous. In Fig. 2f, a linearly varying angle ChG deposition was prepared and drawn so that the onset of NW formation could be validated (Supplementary Note 9). Our Lagrangian framework monitoring the evolution of in-plane instabilities during the draw thus unravels the rationale behind the process presented in this work, driven by the interplay between disjoining and surface tension pressures.

## Achieved geometries

For NWs to be used as building blocks for optics and electronics, a high degree of control over their diameter and spacing is required. Based on our process and fundamental understanding, we now demonstrate our ability to effectively control dewetting periodicity and access additional degrees of freedom in NW array architectures, compared to previous studies. After drawing, a fiber section was prepared to characterize the final geometry of the dewetted film. Figure 3a shows the well-ordered filamentation occurring along the entire cross-section of the drawn film. For $P$ = 10 μm and DR = 60, we achieved meters-long 50 nm-diameter NWs in a single draw (Supplementary Note 10), an very high aspect ratio. The only limit to the number of ChG filaments is given by the size of the drawing tower furnace that limits the preform diameter. For example, for a 30 mm-wide preform and $P$ = 10 μm, one can expect an array of 3000 filaments. Smaller diameters and periods could be attempted at higher DR ($10^2$ to $10^3$ is typical for optical fiber drawing), by successive redrawing[50], or using templates with smaller periods. As explained previously, thermal drawing confers a strong dampening force to longitudinal instabilities[51], allowing in principle for ChG NW diameters in a PEI matrix to reach sub-10 nm dimensions[50].

In nanophotonics, wavelength dimensions are beneficial to tight optical confinement, reaching single-mode guiding regimes, and increasing the proportion of evanescent power of the guided mode for near-field optical sensing[4]. We further demonstrate that by selecting the right deposited ChG film thickness and template period (Fig. 3b), one can tune NWs to both the desired diameter and spacing (Fig. 3c). The different fibers tested thus far span period/diameter ratios between 2 and 6 (Fig. 3d). For equal film thicknesses, various periods can be achieved. By contrast, for a given thickness, a flat film would break up around some mean period, given by the fastest-growing instability wavelength described above in the model, if it breaks up at all. By volume conservation, this would impart the wire diameter.

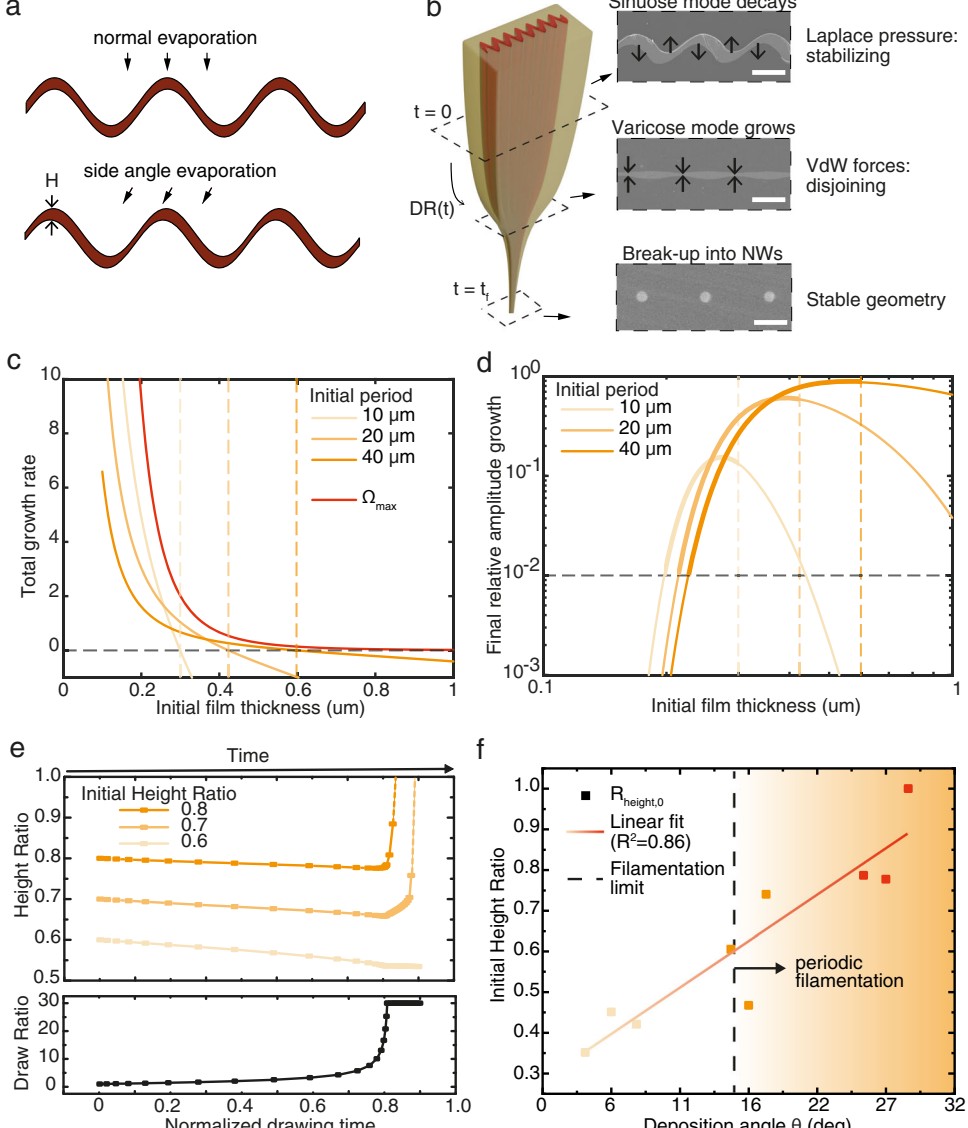

**Fig. 2 | Filamentation models: linear theory, numerical simulation and experimental validation. a** Cross section schematic showing how a thickness fluctuation is obtained by glancing angle deposition on a templated substrate. **b** Lagrangian description of in-fiber templated instabilities. Cross section schematics and SEM cross sections show sinuose/varicose modes evolution and dominant competing pressures in the capillary breakup process. Scale bars from top to bottom are repectively 10 μm, 5 μm and 1 μm. **c** Linearized model of the total instability growth rates during thermal drawing at three experimental template periodicities (termed "initial period") and for the fastest-growing perturbation, for varying deposition thicknesses. Vertical dashed lines show the maximum thickness for which growth rates remain positive (horizontal dashed line). **d** Relative

amplitude growth $R_{growth}$, for varying film thicknesses. The horizontal dashed line shows the limit above which the template instability remains 100 times larger than the fastest growing one. Using vertical dashed lines from (**c**), a range of thicknesses for regular filament formation is estimated for each of the three cases (sections in bold). **e** Time-discretized Lagrangian CFD simulation: (top) For $R_{height,O}$ varying from 0.6 to 0.8, evolution of $R_{height}$ in time evaluated using the complete expression of the VdW potential. $H_O = 0.5$ μm; (bottom) Associated DR as a function of time. Note that it is held constant at 30 during the late stages of the draw to model the fiber's exit from the oven. **f** Linear regression of $R_{height,O}$ as a function of the ChG deposition angle **θ** (see Supplementary Fig. S12), allowing experimentally to determine that for $H_O ~ 500$ nm and $R_{height,O} > 0.6$, NW formation occurs.

---

Therefore, for flat films, the period/diameter ratio would be solely a function of initial film thickness, estimated numerically in Fig. 3d (solid line) with our linearized model[24]. This extra degree of freedom accessed by templating transversal instabilities opens applications in far-field coherent scattering applications, as will be discussed in the applications.

The cladding can also be dissolved for the wires to be used separately (Fig. 3e) or in a bundle (Supplementary Fig. S19). By exposing them, they can be contacted and become electrochemically sensitive to their surrounding environment, as has been done for functionalized NW FET devices[52–54], leveraging the high surface sensitivity of their high aspect ratio. They can also be manipulated into

optical assemblies using optical trapping and optoelectronic tweezers[3]. To increase the density of NWs for optics applications relying on 3D arrangements such as photonic crystals[55] and metamaterials[9], we also show that our process is not limited to a 2D space. By using other template shapes, PVD anisotropy can be used to deposit thicker layers at prescribed locations to achieve sub-period dewetting[34]. We here demonstrate that by evaporating directly a layer onto a template made of rectangular grooves[56] (Supplementary Note 1), several NWs can be obtained for each period (Fig. 3f): a second array of NWs can be seen in a different plane from the main one, enabling "quasi-3D" architectures. Finally, to achieve matrices of filaments truly spanning all dimensions of space, we prepared a preform

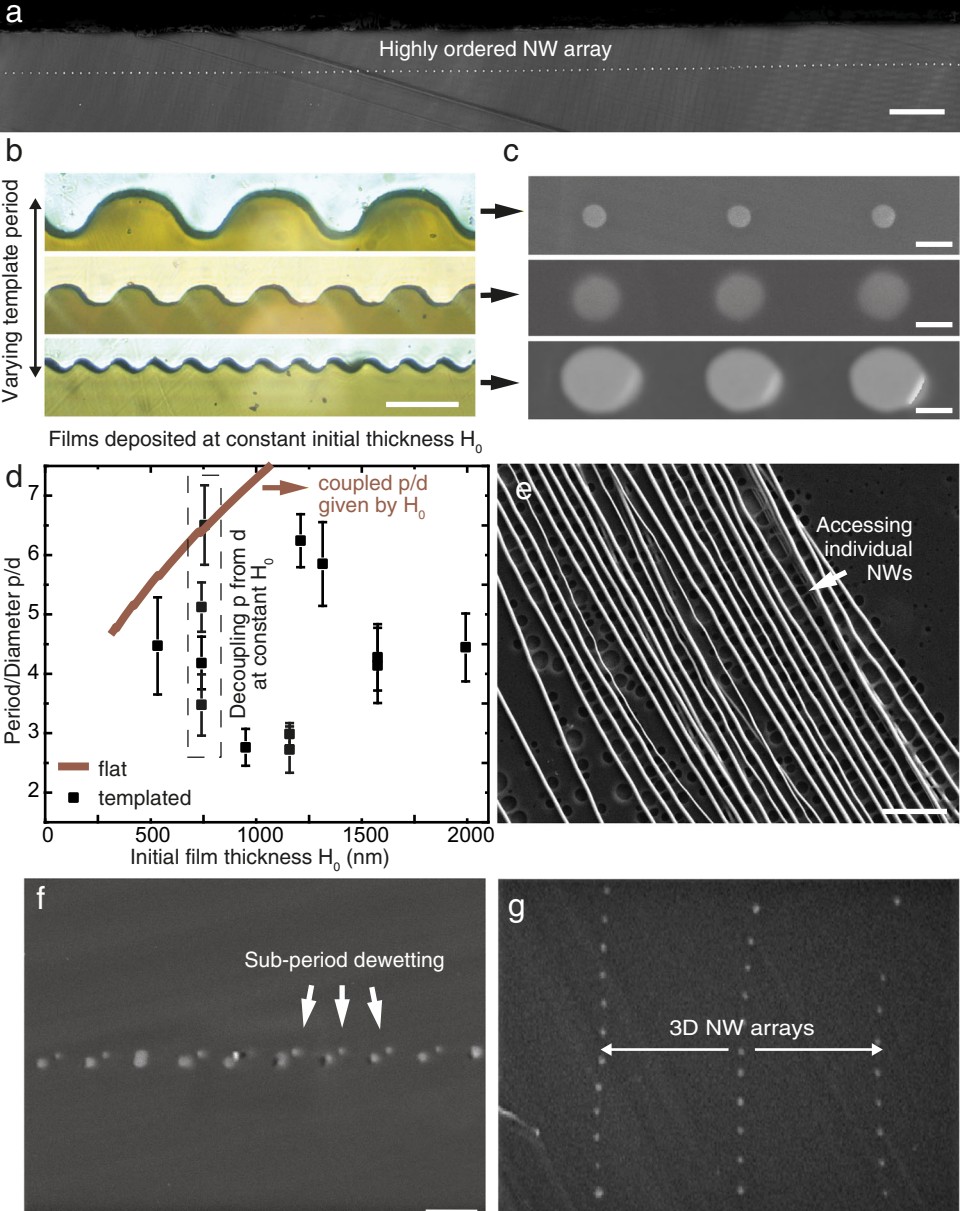

**Fig. 3 | Versatility in geometries and structures. a** SEM picture of long-range ordered filaments in a fiber cross-section. **b** Optical micrographs of cross-sections of the ChG films after deposition onto polymer substrate with different template periods. **c** SEM pictures of the corresponding NW array geometry. **d** Period/diameter ratio for some fibers tested, compared to the flat film case (solid line), plotted against deposited film thickness. In the latter case, only one period/diameter ratio is accessible for a given film thickness. **e** Filaments can be accessed individually when dissolving the cladding. **f** Small satellite NWs can be seen for each larger NW, demonstrating the possibility to achieve sub-period dewetting. **g** Layers can be stacked to make three-dimensional architectures. Scale bars are respectively: **a** 10 μm, **b** 20 μm, **c** 200 μm (top), 100 μm (middle), 50 μm (bottom), **e** 1 μm, **f** 2 μm, **g** 1.5 μm.

consisting of several stacked films. In a single step, they were drawn together, breaking up into multiple arrays of filaments simultaneously. The SEM cross-section in Fig. 3g shows three such layers, paving the way towards 3D NW arrangements.

## Applications

**Diffraction.** The additional degrees of freedom in fiber dewetting enabled by our process call for new applications. In optics, the long-range order of the filament array leads to far-field diffraction, which makes these arrays ultralong flexible gratings, both in transmission (Fig. 4a insert) and reflection. The fiber can be cut into pieces for making numerous individual gratings, at very high throughput and low manufacturing costs: a preform can be drawn within hours into a fiber whose length scales quadratically with DR. A 50 cm-long preform, for

instance, for $DR_f = 100$, yields a 5 km-long fiber, corresponding to 50'000 10 cm-long flexible gratings. Moreover, as aforementioned, by adjusting the template period and deposited thickness, one can not only adjust the grating period but also the wire diameter to best achieve the desired diffraction efficiency at a given wavelength, as will be shown in this section. These parameters are analogous to groove density and groove shape in conventional gratings. To demonstrate this ability, we drew two fibers of similar periods $p$ (1405 nm ± 60 nm), but of different NW diameters $d$ (236 nm and 329 nm). Using a rotating stage (see Methods), we compared the spectra of the first-order diffraction efficiency to the specular one for both fibers. In Fig. 4a, we plotted the intensity ratio $R_{01} = \frac{\eta_{m=1}}{\eta_{m=0}} = \frac{I_{m=1}}{I_{m=0}}$ where $\eta_m$ is the diffraction efficiency of the transmitted order m, and $I_m$ its intensity. Although they diffract at the same angles, the two fibers exhibit different

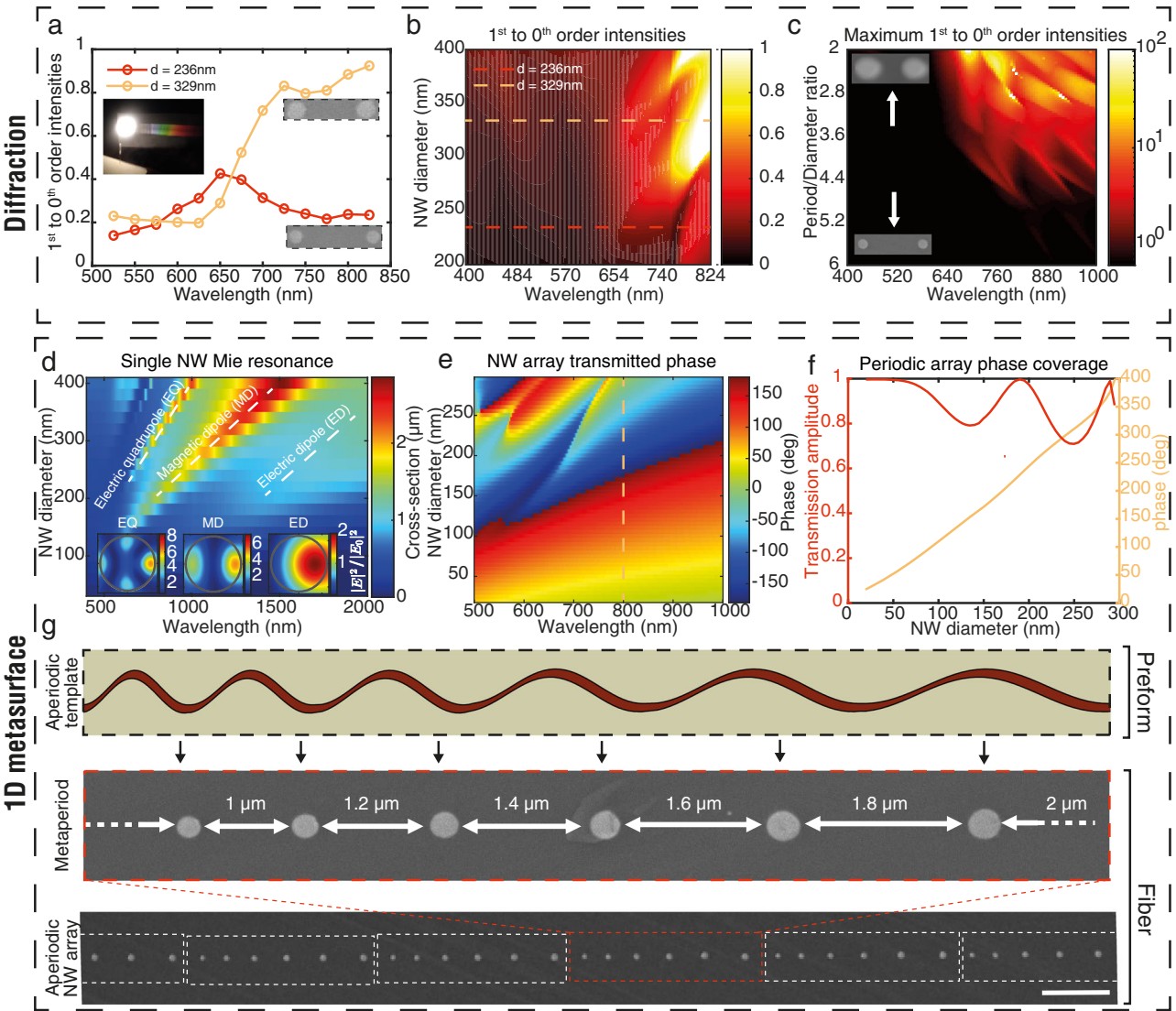

**Fig. 4 | Optical characterization and applications. a** Experimental measurements of $R_{01}$ against incident beam wavelength for two fibers of similar NW array period $p \approx 1.4\,\mu m$, with different NW diameters. **b** From RCWA, $R_{01}$ spectra for various NW diameters $d$ ($p = 1.4\,\mu m$). The two fibers assessed experimentally are represented with dotted lines. **c** For given values of incident beam wavelength $\lambda_i$ and a given ratio period/diameter $p/d$, we vary $p$ and $d$ (respectively in [50 nm, 400 nm] and [$2d$, $6d$]) to find maximal $R_{01}$. A heatmap of $R_{01}$ is plotted showing high diffraction for $\lambda_i > 650$ nm and high grating fill factors (filaments closer to each other). **d** Simulated scattering cross-section spectra of a single ChG NW of varying diameter under a normally incident plane wave in air. Three Mie resonant modes are identified with dashed lines and maps of their normalized field intensity around the wire are plotted as inserts, for a wire diameter of 240 nm. **e** Simulated phase shift spectra in transmission for a 300 nm-period NW array and varying diameters. **f** Normalized transmission amplitude and full phase shift coverage (dashed line in (**e**)) through As₂Se₃ NW arrays of varying diameter, with period $p = 300$ nm, under normally-incident illumination at 800 nm. In figures (**a**–**f**), for clarity, only results for an incident beam polarized parallel to the NWs are shown. **g** Long-range metaperiodicity following the template. The distance between filaments can be tailored individually, paving the way towards one-dimensional metasurfaces.

efficiency spectra between orders. The fiber with thinner NWs shows a dip in $R_{01}$ in the red when the one with thicker NWs peaks. Using rigorous coupled wave analysis (RCWA), we simulated reflected and transmitted efficiencies in all orders, as detailed in Supplementary Note 12. We further explored $R_{01}$ by simulation for a wide range of diameters and incident wavelengths $\lambda_i$. Figure 4b shows an example of the obtained spectra for $d$ varying from 200 to 400 nm, at constant period $p = 1.4\,\mu m$. Dotted lines situate the two fibers assessed experimentally. For a chosen operating wavelength our process enables tuning period and diameter independently so that, for instance, $R_{01}$ can be optimized. To predict how efficient the grating can be at diffracting in the first order, for a given $\frac{p}{d}$ ratio, diameters and periods were swept within ranges obtained experimentally ($d \in$ [50 nm, 400 nm] and $p \in$ [$2d$, $6d$]) to find maximal values for $R_{01}$. Maxima were combined in a heatmap in Fig. 4c. It is noteworthy that high diffraction

is rather achieved for wavelengths larger than 650 nm and filaments as close to each other as possible with respect to the period. This effect can be explained by describing inter and intra-filament modes in a multiple scattering framework[57] which we envision to develop in future work.

**1D Metasurface.** High index contrast ($\frac{n_{As_2Se_3}}{n_{PEI}} > 2$ for $\lambda_i > 650$ nm) and relatively low-loss[58] ($k_{As_2Se_3} < 0.4$ for $\lambda_i > 650$ nm)[34] dielectric inclusions such as ChG NWs can exhibit subwavelength localization of light. Such Mie resonances benefit from lower losses than their plasmonic counterpart[34]. These nanostructures thus offer a way to manipulate light efficiently below the diffraction limit based on their well-defined shapes and dimensions. To illustrate this assertion, we numerically investigated the scattering response of a single ChG NW to a normally incident plane wave using a finite-difference time-domain (FDTD)

software. Figure 4d shows the multipolar scattering response in air for varying wavelengths and NW diameters in the range we achieved experimentally. The first three main Mie resonances[58] are represented with dashed lines. The cross-section normalized field intensity of these modes is represented in inserts. Such high-index inclusions are sub-wavelength building blocks which can be assembled into metasurfaces for wavefront manipulation. As an example, we showed in Fig. 4e the resulting transmitted phase (for transmitted amplitude and details, see Supplementary Note 13) of NWs arranged in a periodic array, using FDTD simulations. We investigated various diameters, under visible/near-IR illumination. Figure 4f is a cut through the map at an 800 nm illumination. At this wavelength, a full $0–2\pi$ phase coverage can be achieved while maintaining high transmission (>75% of incident amplitude), enabling wavefront control[59]. Based on this principle, flat focusing gratings have been proposed[60]. We further showed the possibility to tune independently every single spacing (and diameter) between two consecutive NWs. As an example, a substrate was templated with supercells consisting of linearly increasing periods $P$ from 20 μm to 40 μm, in 4 μm increments. This supercell was repeated over the entire imprint to template the ChG film. The resulting filamentation still followed well the template aperiodicity, leading to a "meta-arrangement" of NWs. An SEM cross-section of the filaments in the fiber cladding is shown in Fig. 4g. Such geometrical arrangements in transmitting arrays are exploited as subwavelength-thick beam deflector devices[61]. Thus, we believe the proposed filamentation process demonstrated here opens a pathway with minimal steps for the manufacturing at large scale of in-fiber 1D photonic metasurfaces and flat optics devices.

## Discussion

Based on a model for anisotropic instabilities in a viscous sheet under stretching, we proposed a process to obtain large-scale meters-long nanowires with controllable spacing in a single fiber drawing. This was achieved by templating the fiber preform, followed by a well-chosen glazing incident deposition, to induce strong film thickness fluctuations. After drawing, the resulting ultralong NWs exhibit a high degree of order. Film dewetting dynamics were described as two competing forces, namely surface tension to maintain stability and Van der Waals forces to drive the instability. We used a linearized model approximation to describe under which conditions the selected thickness perturbation wavelength of templated thickness dominates film break-up. We further developed a script to predict the evolution of any film geometry throughout thermal drawing by running iteratively scaled CFD simulations. As opposed to flat films, we then demonstrated that our process enables tuning independently NW diameter and spacing, based on the initial template period and deposited thickness. Several arrays can be stacked to make three-dimensional structures. Chalcogenide arrays were characterized for optical diffraction. The ability to tune independently NW diameter and spacing allows tuning of diffraction efficiencies. Finally, since the template for deposition can be designed to an arbitrary periodicity, we hinted at the possibility to obtain metasurfaces in multimaterial fibers by self-assembly during fiber drawing. This simple high throughput process opens a pathway to manufacture cost-efficient ultralong tunable NW arrays for nanophotonics and optoelectronic applications requiring long-range order over several meters.

## Methods

### PDMS micro-imprinting

The PDMS mold is replicated from a silicon one obtained by dry reactive etching and photolithography at EPFL's Center for Microtechnology clean room. To obtain the Silicon master template, a standard silicon wafer is coated with 1 μm AZ ECI 3007 photoresist, then exposed in a Heidelberg VPG 200 laser pattern generator, and dry etched in an Alcatel AMS 200 SE with fluorine chemistry. The PDMS

mold is finally used in a NanoImprint EHN-3250 thermal nanoimprinter to hot-emboss a thermoplastic (PEI or PSU) film. Imprints are checked with optical profilometry. In this work, the mold's period $P$ was chosen to be 10, 20 or 40 μm.

### Chalcogenide deposition

ChGs ($As_2Se_3$ or Se) are deposited in a custom-built thermal evaporator comprising an Oerlikon UNIVEX 250 chamber. Current heats up glass chunks in a boat under high vacuum. The imprinted/embossed films are taped onto a stage rotating for deposition homogeneity. The setup and average deposition angle are detailed in Supplementary Note 2. This angle is controlled by placing the sample on custom-made pyramidal structures. The deposited thickness is measured by a quartz crystal microbalance, calibrated by measuring the depth of a scratch in the ChG film on a Si fragment by optical profilometry.

### Preform consolidation and thermal drawing

To encapsulate the ChG film after deposition into the fiber preform, it was placed between a thermoplastic (PEI or PSU) plate and a flat 100 μm-thick film. The stack was placed in a Meyer hot press at 250 °C for 10 min. This preform "consolidation" time is kept to a minimum to limit isotropic annealing to the minimum required for encapsulation of the ChG. The fiber was then drawn in a custom-built drawing tower. The preform was fed into a furnace, typically at 1 mm/mn. The final draw ratio was usually 30 (drawing at 0.9 m/mn), except for achieving 50 nm-wide filaments, for which a draw ratio as high as 60 was reached. The furnace temperatures were set to 210, 360 and 130 °C for the top, center and bottom parts respectively. The actual air temperature inside the oven is thought to be about 50 °C lower. Drawing faster (feed speed = 2 mm/mn for the same draw ratio) was also successfully attempted using -10 °C higher furnace temperatures.

### Fiber cross-sections characterization

When the NW array gets small, optical microscopy is hindered by the diffraction limit. In transmission, only interference patterns can be seen, although this gives a first idea of the regularity of the filamentation. A clean cross-section of the fiber is therefore required to precisely understand the final ChG film geometry at the nanoscale within the polymer cladding. The best results were obtained using a Gatan ILION II ion polisher. Reasonably good surfaces could also be prepared faster using a Leica UC 7 ultramicrotome. The cross-sections were then carbon coated (10 nm) for observation in a Zeiss GeminiSEM field emission SEM equipped with a GEMINI II column operating at 3.0 kV with a 30 μm aperture. For small NWs at high magnification, a lower acceleration voltage (<1 kV) had to be used to avoid sample degradation. One could also dissolve the fiber cladding to analyze the wires, but doing so destroys their periodic spacing and encapsulation. Diameter and spacing distributions were computed from image analysis of the SEM pictures in ImageJ.

### Diffraction setup

Diffraction efficiency was measured using a PM100D Thorlabs Digital Power Meter with the S120C Photodiode Power Sensor on a rotating stage to follow the first transmitted order at different angles. The incident beam was prepared from a SuperK EXTREME supercontinuum laser from NKT photonics through a 100 μm-wide slit restricting the beam to the NW array area in the fibers tested. The laser was used at 10% of its maximum power with a 10 nm bandwidth. See Supplementary Note 11 for the experimental setup. Although laser power depends on the wavelength, by measuring the first order diffraction relatively to the specular, such power variations are cancelled out. The power meter photodiode's distance to the fiber was adjusted to encompass whole transmitted orders despite some angular dispersion due to the NW diameter dispersion (see distribution characterization in

Supplementary Note 10), but far enough not to measure several orders at the same time.

## Data availability
The datasets generated during the current study are available from the corresponding author on request.

## Code availability
The Matlab codes used for the two filamentation models are available from the corresponding author on request. The main parameters used in the scripts are described in Supplementary Notes 7 and 8 for the linearized model and the CFD model, respectively. The RETICOLO software used for grating analysis is available at https://doi.org/10.48550/arXiv.2101.00901.

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

## Acknowledgements

F.S. thanks the ERC-2018-PoC funding scheme (Project SENFLEX – 842349), as well as the basic laboratory funding from EPFL. D.D. acknowledges funding by the National Program in China and startup in Fudan University.

## Author contributions

F.S. conceived the idea with inputs from W.E. and L.M.M. F.S. supervised the project. W.E. and L.M.M. planned the experiments. W.E. performed the experiments and analyzed the data with inputs from F.S and L.M.M. L.M.M. performed the iteratively scaled CFD simulations. B.X. and D.D. conceived the linearized model for perturbation growth and B.X. ran the related numerical simulations. W.E. and P.P. performed the RCWA analysis for diffraction simulation. P.P. ran the FDTD software simulations. W.E. made the figures with inputs from all co-authors. W.E. and L.M.M. wrote a draft of the manuscript. W.E. organized and compiled the first version of the manuscript with inputs from all co-authors. W.E., F.S. and D.D. revised the final versions of the manuscript.

## Competing interests

The authors declare no competing interests.
