## [Peer Review File · Nature Communications]

Controlled filamentation instability as a scalable fabrication approach to flexible metamaterialsREVIEWER COMMENTS

Reviewer #1 (Remarks to the Author):

This paper presents a method for the production of nanowire arrays, and demonstrates its utility for the production of materials with both diameter and spacing control. The manuscript is well written and the technique will be of interest to those working in the field.

Comments:

Publication with minor corrections/additions is recommended

a few specific comments:

Although it is rolled into the LaPlace pressure, surface tension is a concept that is more familiar to the reader, and might make the discussion more accessible.

The split of fig 1b into two parts (i & ii) was not as clear as it could be. Perhaps separate into b & c, or use brackets to indicate which images belong to each part. The caption made it possible to parse, but the figure itself was confusing.

The use of a CH franc as a scale bar is whimsical, but not useful to everyone. Please add a conventional scale bar to Fig 1c, or consider removing the panel. There is no indication of the particulars of the fiber shown, and the human hand in panel d gives a general idea of scale.

Line 137 "for the initial textured film" only has clear meaning if the ratio of thicknesses and period are given, as in the subsequent discussion. Perhaps "for a typical textured film" the ratio of amplitudes is 100??

Line 194

"193 As explained previously, thermal drawing confers a strong dampening force to longitudinal 194 instabilities⁴⁸, allowing in principle for ChG NW diameters in a PEI matrix to reach sub-10 nm dimensions⁴⁷. In 195 nanophotonics, wavelength dimensions are beneficial to tight optical confinement, reaching single-mode guiding 196 regimes, and increasing the proportion of evanescent power of the guided mode for near-field optical sensing⁴."

This was confusing – led me to believe that some applications for *very short* wavelengths would be discussed. A paragraph break before "In nanophotonics..." would fix this.

Please correct the italic text in this paragraph:

Supplementary Note 3

"While consolidation implies a longer nucleation and growth of hole stage, dip-coating of thick encapsulating layers often leaves significant residual stress within the encapsulating material, which may lead to delamination. In the rest of this paper, we resort solely to the use of the consolidation technique, which is the most practical. Furthermore, we show in Supplementary *Note 4* that when annealing the sample for consolidation, before the top layer softens and encapsulates the ChG film, open-air reflow can be beneficial to increasing templated film thickness fluctuations. "

General – response appreciated, not required...

The optical simulations (as expected) show little effect for wire dimensions as small as 10 nm, presumably an effective medium description is appropriate (not needed here). However, plasma resonance effects from metallic nanoparticles of that size are well documented, and broader spectral absorption should result from nanowires (with polarization dependence?) – Please add a short discussion of the implications of your method for plasmonics, or elaborate on why the method cannot be extended to metals.

The work in Supplementary note 7 is a good introduction to the importance of interface effects,

but suggestions of what might be expected with use of different polymer hosts, or hosts with interfacial layers would be of interest. Would a phase change (Sn melting, e.g.) violate the assumptions in your modelling?

Related to this, a discussion of the possibilities/limitations of the technique – e.g. the relative interface interaction strength vs surface tension needed to form nanowires would be most helpful, along with anticipated effects of phase transitions – are films that will liquify a non-starter? What is the glass transition temperature of the chalcogenide materials used? Some examples of other materials with the relative viscosities needed would be of interest.

Reviewer #2 (Remarks to the Author):

In the current manuscript the authors report on a scalable process (fiber drawing) to generate encapsulated flexible nanowire arrays (composition: selenium-based amorphous semiconductors) with improved aspect ratios, size tunability and periodicity.

The real novelty is to control well-ordered extremely long nanowire self-assembly architectures via the filamentation of a textured thin film under anisotropic stretching. The method involves coupling soft lithography, angled thermal evaporation, and thermal drawing.

Potential applications of the exposed method are proposed (tuning diffraction to a given grating period, possibility of manufacturing macroscopic meta-grating superstructures for nanophotonic applications).

The paper is very well written and organized.

It includes a strong set of experimental data supported by numerical simulation and detailed supplementary results, making the manuscript innovative and of potential interest for the specialty optical fiber community.

In conclusion, the manuscript deserves publication in "Nature Communications".

Minor changes:

- Some inconsistencies with text alignment in "Supplementary section"

Controlled filamentation instability as a scalable fabrication approach to flexible metamaterials

Manuscript NCOMMS-22-21958-T

Point-by-point response to the reviewers

Reviewer #1

This paper presents a method for the production of nanowire arrays, and demonstrates its utility for the production of materials with both diameter and spacing control. The manuscript is well written and the technique will be of interest to those working in the field.

Comments:

Publication with minor corrections/additions is recommended

Thank you for your support and for recommending our manuscript for publication after minor changes.

a few specific comments:

Although it is rolled into the LaPlace pressure, surface tension is a concept that is more familiar to the reader, and might make the discussion more accessible.

Thank you for your comment. We replaced “Laplace pressure” by “surface tension” in the main text.

The split of fig 1b into two parts (i & ii) was not as clear as it could be. Perhaps separate into b & c, or use brackets to indicate which images belong to each part. The caption made it possible to parse, but the figure itself was confusing.

Thank you for the suggestion. We hope the new layout is clearer. The caption and references to Fig 1b and c have been updated accordingly.

The use of a CH franc as a scale bar is whimsical, but not useful to everyone. Please add a conventional scale bar to Fig 1c, or consider removing the panel. There is no indication of the particulars of the fiber shown, and the human hand in panel d gives a general idea of scale.

Thanks for this suggestion! We have added a scale bar to Fig 1c.

Line 137 "for the initial textured film" only has clear meaning if the ratio of thicknesses and period are given, as in the subsequent discussion. Perhaps "for a typical textured film" the ratio of amplitudes is 100??

Thanks for this suggestion! We have clarified this sentence, as “for a typical textured film”.

Line 194

*"193 As explained previously, thermal drawing confers a strong dampening force to longitudinal
194 instabilities⁴⁸, allowing in principle for ChG NW diameters in a PEI matrix to reach sub-10*

nm dimensions 47. In nanophotonics, wavelength dimensions are beneficial to tight optical confinement, reaching single-mode guiding regimes, and increasing the proportion of evanescent power of the guided mode for near-field optical sensing⁴."

This was confusing – led me to believe that some applications for very short wavelengths would be discussed. A paragraph break before "In nanophotonics..." would fix this.

Thank you! We have inserted the paragraph as below:

"As explained previously, thermal drawing confers a strong dampening force to longitudinal instabilities⁴⁸, allowing in principle for ChG NW diameters in a PEI matrix to reach sub-10 nm dimensions⁴⁷.

In nanophotonics, wavelength dimensions are beneficial to tight optical confinement, reaching single-mode guiding regimes, and increasing the proportion of evanescent power of the guided mode for near-field optical sensing⁴."

Please correct the italic text in this paragraph:

Supplementary Note 3

"While consolidation implies a longer nucleation and growth of hole stage, dip-coating of thick encapsulating layers often leaves significant residual stress within the encapsulating material, which may lead to delamination. In the rest of this paper, we resort solely to the use of the consolidation technique, which is the most practical. Furthermore, we show in Supplementary Note 4Error! Reference source not found. that when annealing the sample for consolidation, before the top layer softens and encapsulates the ChG film, open-air reflow can be beneficial to increasing templated film thickness fluctuations. "

Thank you. We have corrected this typo by including "Supplementary Note 4".

General – response appreciated, not required...

The optical simulations (as expected) show little effect for wire dimensions as small as 10 nm, presumably an effective medium description is appropriate (not needed here). However, plasma resonance effects from metallic nanoparticles of that size are well documented, and broader spectral absorption should result from nanowires (with polarization dependence?) – Please add a short discussion of the implications of your method for plasmonics, or elaborate on why the method cannot be extended to metals.

Plasmonic nanowire arrays are indeed of interest for many applications such as waveguiding¹, sensing² or selective absorption/reflection³. We believe our fabrication method can be extended to plasmonics as long as i) metals can be thermally drawn in a liquid form in an appropriate cladding and ii) instabilities do not grow too fast in the longitudinal axis (along the fiber).

We see no hindrance to i) as such structures have already been obtained by the stack-and-draw method³⁻⁵. The conditions under which ii) is fulfilled and metal nanowire arrays can be obtained by the stack-and-draw method without capillary break-up have already been studied^{6,7}. We also detailed why (metal) wire longitudinal instabilities are suppressed by axial stretching^{8,9}. Under such conditions, our process allows NW formation in a single draw by enhancing film thickness

transverse instabilities. Our method would only be limited to the subset of available metals which can be thermally evaporated. Furthermore, we envision that our work with metallic glass thermal drawing¹⁰ opens exciting opportunities where metal nanowires could be obtained in a viscous regime.

The work in Supplementary note 7 is a good introduction to the importance of interface effects, but suggestions of what might be expected with use of different polymer hosts, or hosts with interfacial layers would be of interest. Would a phase change (Sn melting, e.g.) violate the assumptions in your modelling?

Related to this, a discussion of the possibilities/limitations of the technique – e.g. the relative interface interaction strength vs surface tension needed to form nanowires would be most helpful, along with anticipated effects of phase transitions – are films that will liquify a non-starter? What is the glass transition temperature of the chalcogenide materials used? Some examples of other materials with the relative viscosities needed would be of interest.

Thank you for the comments on the modelling!

Physically, the underlying mechanism is based on the Van der Waals force, and the theory is addressing this force or the associated disjoining pressure to drive the film instability and final filamentation. From linear analysis, the growth rate depends on the wavelength, and the amplitude for a given wavelength is determined by both the growth rate and the prescribed amplitude at a prescribed wavelength of the textured film. In such a fashion, we can control the filamentation instability, in terms of the dominant instability wavelength or the final period of filaments.

Regarding the relative interface interaction strength vs surface tension for the surface tension to form nanowires, one can obtain the quantitative understanding of instability growth rate dependent on these parameters, such as their ratio. From the linear stability analysis for a sandwiched film model in our previous theoretical work⁸, the growth rate increases with the Van der Waals force for a given surface tension, as shown from the figure as below, which is Fig. 3 in reference [8].

FIG. 3. Instability of a sandwiched sheet due to van der Waals force. (a) Growth rate Ω versus wave number K ($\alpha = 100$, $\eta = 1$); (b) Maximum growth rate Ω_m (solid line) and its corresponding wave number K_m (dashed line) dependent on dimensionless ratio α of van der Waals force to interfacial tension ($\eta = 1$).

For other materials, the effects of various physical parameters on the dewetting instability have been extensively investigated in our previous theoretical work⁸, including Van der Waals forces, viscosities, surface tensions, and thicknesses. Specifically, the maximum growth rates for various material combinations are listed below, which is Table 1 in reference [8], and more details can be found in section “Application of fiber drawing”⁸.

TABLE I. Controlling instability or the perturbation growth rates through designing various structures for application of fiber drawing (Hamaker constant of thin sheets $A_j \sim 10^{-17}$ J, interfacial tension of interfaces $\gamma^{(i)} \sim 10^{-1}$ N m⁻¹).

Structure	Viscosities (Pa s)	Thicknesses (nm)	Growth rate ω_m (s ⁻¹)
PES/As ₂ Se ₃ /PES	10 ⁵ /10 ⁵ /10 ⁵	∞ /10/ ∞	0.703
PSU/Se/PSU	10 ⁵ /1/10 ⁵	∞ /80/ ∞	0.184
PES/As ₂ Se ₃ /PES	10 ⁵ /10 ⁵ /10 ⁵	∞ /50/ ∞	1.33×10^{-3}
PES/As ₂ Se ₃ /Se	10 ⁵ /10 ⁵ /1	∞ /50/ ∞	2.56×10^{-3}
PES/As ₂ Se ₃ /Se/PES	10 ⁵ /10 ⁵ /1/10 ⁵	∞ /50/50/ ∞	0.952
PSU/Se/As ₂ Se ₃ /PSU	10 ⁵ /1/10 ⁵ /10 ⁵	∞ /30/30/ ∞	5.557
PSU/Se/As ₂ Se ₃ /PSU	10 ⁵ /1/10 ⁵ /10 ⁵	∞ /30 H_{ct} /30/ ∞	0.0191
Se/As ₂ Se ₃ /PSU	1/10 ⁵ /10 ⁵	∞ /30/ ∞	0.0191

Based on our previous model^{8,9}, we carefully address the effects of drawing on the filamentation instability. The wavelength of the varicosity corresponding to the maximum instability increases indefinitely as the ratio of viscosities between the film and surrounding cladding tends to infinity or to zero. In such limit cases, our templated wavelength wouldn’t dominate the breakup mode in terms of the final nanowire period. Our process therefore works best for cases where film/cladding viscosities are comparable or not too dissimilar. However, even for a liquid phase in a viscous cladding, if templated instabilities start at a sufficiently high amplitude (high deposition angle), the prescribed mode may still be able to determine the final breakup mode.

For a predictive and quantitative answer, as presented in Supplementary Note 7, our model would adapt to simulating various polymer hosts and interfacial layers by varying the corresponding viscosities, Van der Waals coefficients and surface tensions. The physical parameters vary in a complex manner along the drawing direction with the temperature profile inside the furnace and the scaling resulting from anisotropic stretching, which Computational Fluid Dynamics can solve numerically in the general case. Here the phase change is characterized by a temperature-dependent viscosity.

References for the answer to reviewer #1

1. Hou, J. *et al.* Metallic mode confinement in microstructured fibres. *Opt. Express, OE* **16**, 5983–5990 (2008).

2. Guo, X., Ying, Y. & Tong, L. Photonic Nanowires: From Subwavelength Waveguides to Optical Sensors. *Acc. Chem. Res.* **47**, 656–666 (2014).
3. Tuniz, A. *et al.* Drawn metamaterials with plasmonic response at terahertz frequencies. *Appl. Phys. Lett.* **96**, 191101 (2010).
4. Tuniz, A. *et al.* Metamaterial fibres for subdiffraction imaging and focusing at terahertz frequencies over optically long distances. *Nature Communications* **4**, 2706 (2013).
5. Tuniz, A. *et al.* Stacked-and-drawn metamaterials with magnetic resonances in the terahertz range. *Opt. Express, OE* **19**, 16480–16490 (2011).
6. Xue, S., Barton, G. W., Fleming, S. & Argyros, A. Analysis of Capillary Instability in Metamaterials Fabrication Using Fiber Drawing Technology. *J. Lightwave Technol.* **35**, 2167–2174 (2017).
7. Zhang, X., Ma, Z., Yuan, Z.-Y. & Su, M. Mass-Productions of Vertically Aligned Extremely Long Metallic Micro/Nanowires Using Fiber Drawing Nanomanufacturing. *Advanced Materials* **20**, 1310–1314 (2008).
8. Xu, B. *et al.* Filament formation via the instability of a stretching viscous sheet: Physical mechanism, linear theory, and fiber applications. *Phys. Rev. Fluids* **4**, 073902 (2019).
9. Xu, B. & Deng, D. Linear analysis of dewetting instability in multilayer planar sheets for composite nanostructures. *Phys. Rev. Fluids* **5**, 083904 (2020).
10. Yan, W. *et al.* Structured nanoscale metallic glass fibres with extreme aspect ratios. *Nat. Nanotechnol.* **15**, 875–882 (2020).

Reviewer #2

In the current manuscript the authors report on a scalable process (fiber drawing) to generate encapsulated flexible nanowire arrays (composition: selenium-based amorphous semiconductors) with improved aspect ratios, size tunability and periodicity.

The real novelty is to control well-ordered extremely long nanowire self-assembly architectures via the filamentation of a textured thin film under anisotropic stretching. The method involves coupling soft lithography, angled thermal evaporation, and thermal drawing.

Potential applications of the exposed method are proposed (tuning diffraction to a given grating period, possibility of manufacturing macroscopic meta-grating superstructures for nanophotonic applications).

The paper is very well written and organized. It includes a strong set of experimental data supported by numerical simulation and detailed supplementary results, making the manuscript innovative and of potential interest for the specialty optical fiber community.

In conclusion, the manuscript deserves publication in “Nature Communications”.

Thank you for recommending that the manuscript deserves publication in Nature Communications”.

Minor changes:

- *Some inconsistencies with text alignment in “Supplementary section”*

Thank you for your careful reading! Indeed, some paragraph first line indentation widths were different, and we have corrected these alignments to be consistent.

REVIEWERS' COMMENTS

Reviewer #1 (Remarks to the Author):

Nice work; ready for publication

Reviewer #2 (Remarks to the Author):

The manuscript now deserves publication in Nature Communication.